# AGS-v PLUS, a Mosquito Salivary Peptide Vaccine, Modulates the Response to *Aedes* Mosquito Bites in Humans

**DOI:** 10.3390/vaccines13101026

**Published:** 2025-09-30

**Authors:** Liam Barningham, Ian M. Carr, Siân Jossi, Megan Cole, Aiyana Ponce, Mara Short, Claudio Meneses, Joshua R. Lacsina, Jesus G. Valenzuela, Fabiano Oliveira, Matthew B. Laurens, DeAnna J. Friedman-Klabanoff, Olga Pleguezuelos, Lucy F. Stead, Clive S. McKimmie

**Affiliations:** 1School of Medicine, University of Leeds, Welcome Trust Brenner Building, St. James’ University Hospital, Beckett Street, Leeds LS9 7TF, UK; 2Virus Host Interaction Team, Skin Research Centre, Hull York Medical School, University of York, York YO10 5DD, UK; 3Heyford Park Innovation Centre, ConserV Bioscience Ltd., 77 Heyford Park, Bicester OX25 5HD, UK; 4Laboratory of Malaria and Vector Research, National Institute of Allergy and Infectious Diseases, National Institutes of Health, Rockville, MD 20852, USA; 5Department of Entomology, College of Agriculture, Life, and Environmental Sciences, University of Arizona, Tucson, AZ 85721-0036, USA; 6Center for Vaccine Development and Global Health, University of Maryland School of Medicine, 685 W Baltimore St Room 480, Baltimore, MD 21201-1509, USA

**Keywords:** arbovirus, vaccine, skin, mosquito, RNA-seq, saliva, AGS-v PLUS, human, phase I trial, transcriptomics

## Abstract

Background: The global health burden of mosquito-borne viruses, including dengue, yellow fever, Zika, and chikungunya, is rising due to climate change and globalisation, which favour mosquito habitat expansion. The genetic diversity of these viruses complicates the development of virus-specific vaccines or antivirals, highlighting the need for pan-viral strategies. As the common vector for these pathogens, mosquitoes and specifically their salivary proteins represent a promising target for such interventions. Mosquito saliva, secreted into the skin during biting, has immunomodulatory effects that can enhance host susceptibility to infection, but these mechanisms are not well defined in humans. Methods: The objective of this study was to determine whether AGS-v PLUS, a vaccine targeting mosquito salivary antigens, could modulate the human skin immune response to mosquito biting and potentially promote antiviral bystander immunity. In a Phase I trial, healthy volunteers were vaccinated with AGS-v PLUS (with or without adjuvant) or placebo, and three weeks later, they were exposed to bites from *Aedes albopictus* and *Aedes aegypti* mosquitoes. Skin biopsies from bitten and unbitten sites were analysed by transcriptomic profiling. Results: In placebo recipients, mosquito biting elicited a marked adaptive immune response at 48 h, characterised by CD4^+^ Th1 and CD8^+^ T cell signatures and leukocyte recruitment. While responses to *Ae. aegypti* and *Ae. albopictus* bites were broadly similar, those to *Ae. albopictus* were stronger. Vaccination with AGS-v PLUS, particularly with adjuvant, enhanced Th1 and CD8^+^ T cell-associated gene expression while suppressing pathways linked to neutrophilic inflammation and epithelial stress, which together may provide enhanced antiviral capacity. Conclusions: These findings demonstrate that targeting the host response to mosquito saliva via vaccination can reprogram the skin’s immune response to mosquito bites, supporting a novel and broadly applicable pan-viral strategy to mitigate the impact of arboviral diseases.

## 1. Introduction

The threat posed by mosquito-borne viruses (arboviruses) is increasing as climate change intensifies and globalisation facilitates the spread of mosquitoes to new geographic areas [1]. Outbreaks of disease caused by arboviruses, such as dengue, Zika and chikungunya, are increasing in both frequency and severity [2]. These viruses are genetically heterogeneous and numerous, making it challenging to develop virus-specific medicines for them all. Some arboviruses have currently a low incidence rate, and development of virus-specific vaccines is not cost-effective. Consequently, the identification of pan-viral targets which could putatively be used against multiple arboviruses is highly attractive [3,4].

One such pan-viral component of arboviral infection is the saliva deposited by biting arthropods such as mosquitoes [5,6,7,8,9,10,11,12,13]. Infection with mosquito-borne viruses occurs during probing of the skin, as the mosquito seeks to obtain a blood meal. During probing, as mosquitoes search for a suitable capillary to feed from, an infected mosquito punctures the skin and deposits the virus together with saliva, primarily in the dermis. Importantly, the saliva has multiple biologically potent effects on host skin that facilitate mosquito feeding [14,15,16]. Aspects of the host response to this saliva can counterintuitively enhance host susceptibility to the virus [8,17,18,19,20,21,22,23,24]. A common misconception is that biting induces an allergic response in skin in nature, mainly due to rapid onset of oedema that follows a bite. However, pharmacologically active salivary components, such as sialokinin, may instead be responsible for many of these features, including bite swelling [16,18]. Although mouse models have been used successfully to define this inflammatory response to mosquito biting, far less work has been done to study human skin’s immune responses to mosquito biting. As such, there is a significant gap in our understanding of whether mouse-derived data are of relevance to human arbovirus infection. The most authoritative study to date assessing human skin response to mosquito biting demonstrated that the early response to biting is dominated by neutrophil influx, while both Th1 and Th2 T cell responses are activated by 48 h [25]. As such, mosquito bites represent a distinct inflammatory niche.

Vaccines that target specific components of mosquito saliva to modulate immune responses to mosquito biting to become antiviral, as opposed to pro-viral, could provide bystander resistance to infection against multiple genetically distinct viruses [5]. This is particularly relevant for dengue, yellow fever, Zika and chikungunya viruses, as they are all transmitted by just one genus of mosquito, the *Aedes* mosquito. One such vaccine is AGS-v PLUS, which was originally designed to target highly conserved regions of salivary factors from the malarial mosquito vector *Anopheles* and predicted to be immunogenic by using a proprietary algorithm that is able to determine regions that could bind to MHC-I receptors and induce either tolerance or Th1 responses. AGS-v contains four short synthetic peptides that originate from *Anopheles* salivary proteins. Peptide AGS-6 originates from salivary gland 7-like protein, AGS-30 and 31 originate from salivary gland 1-like 3 protein, and AGS-35 originates from the salivary gland 7 protein. The peptide mix was adjuvanted in Montanide ISA-51 (Seppic) and induced strong IFN-γ and antibody responses in mice. A Phase 1 study showed that AGS-v was safe and boosted IgG and IFN-γ responses [9]. AGS-v PLUS adds a fifth peptide, AGS-20, which originates from PRS-1 protein and is found in various mosquito species including *Aedes* genera mosquitoes. As such, AGS-v PLUS contains five synthetic peptides derived from conserved regions of mosquito salivary proteins, designed to induce immune responses that counteract the immunomodulatory effects of mosquito saliva during blood feeding. Preclinical data showed strong IgG titres in mice, while a Phase 1 trial [12] confirmed AGS-v PLUS’s safety and immunogenicity in a cohort of individuals from an area with established *Aedes* populations [9].

In the AGS-v PLUS trial, volunteers were vaccinated with AGS-v PLUS and then exposed to biting mosquitoes. Skin response to biting was assessed by taking a biopsy of the bite at 48 h post-biting. In this study, we make use of this unique human cohort to define human skin response to mosquito biting at the gene expression level with an unprecedented scope. By first characterising the baseline skin response to mosquito biting in placebo participants, we were able to isolate vaccine-specific effects. We define differences and commonalities in inflammatory responses to biting by the two main mosquito vectors of arboviruses, *Ae. aegypti* and *Ae. albopictus*. Crucially, we then assess whether this host inflammatory response is modulated by prior vaccination with AGS-v PLUS. We show that *Aedes*-bitten skin had thousands of genes upregulated by 48 h post-biting, the most significantly and highly upregulated being those encoding chemokines, pro-inflammatory cytokines, innate immune signalling, and some T cell responses. Immune deconvolution of the transcriptome data was used to define leukocyte influx into skin, demonstrating significant increases in immune cells post-biting. Crucially, an antiviral immune signature in response to mosquito saliva was enhanced in those individuals that had received prior adjuvanted AGS-v PLUS vaccination, which could function in an antiviral capacity during infection. Immune deconvolution also identified a signature of leukocyte influx into skin with mosquito biting, which was enhanced for natural killer cells and dendritic cells in individuals vaccinated with adjuvanted AGS-v PLUS.

## 2. Methods

### 2.1. Trial Design and Participants

The trial was a randomized, double-blind, placebo-controlled Phase 1 study of AGS-v PLUS administered on Days 1 and 22 at the University of Maryland School of Medicine’s Center for Vaccine Development and Global Health (CVD) in Baltimore, MD, USA. Healthy adults aged 18–50 years old, inclusive, with a body mass index of 18–40, inclusive, who agreed to use effective contraception (as defined in the protocol) from four weeks before enrolment until 12 weeks after second vaccination and had no history of previous severe allergic reaction, recent immunosuppression, ongoing chronic skin condition other than mild eczema, or other condition that would preclude ability to participate were enrolled. Participants were recruited and enrolled without regard for sex, and we recorded participant sex based on self-report. The full trial protocol, including complete inclusion/exclusion criteria, has previously been described [9]. Crucially, participants represented an equal mix of white and black/African American descent, and 10% were of Asian descent, as self-reported.

### 2.2. Ethics

Written informed consent was obtained from all participants. The study protocol (HP-00076625, FWA00007145) was approved by the University of Maryland Institutional Review Board and was conducted in accordance with the International Conference on Harmonization of Good Clinical Practices and the Declaration of Helsinki. Safety oversight was provided by the NIAID Intramural Data and Safety Monitoring Board. CVD investigators directed the clinical trial and conducted safety assessments. The trial is registered under NCT04009824.

### 2.3. Procedures

AGS-v PLUS was manufactured under Good Manufacturing Practice; peptides were manufactured by Corden Pharma (Liestal, Switzerland) and then formulated into the final drug product by Corden Pharma (Caponago, Italy). Lyophilised vaccine was reconstituted before injection, resulting in a solution containing 50 nmol of each peptide. The placebo group received 0.5 mL sterile saline placebo. For non-adjuvanted AGS-v PLUS, lyophilized AGS-v PLUS was reconstituted with 0.5 mL sterile water for injection (WFI). For the AGS-v PLUS/ISA-51 doses, each AGS-v PLUS vial was emulsified in 0.25 mL Montanide ISA-51 (Seppic, Castres, France) and 0.25 mL WFI to total 0.5 mL. AGS-v PLUS/Alhydrogel doses were prepared by mixing AGS-v PLUS in 0.12 mL Alhydrogel (Sergeant Adjuvants, Clifton, NJ, USA) and 0.38 mL sterile saline to total 0.5 mL. Participants received 0.5 mL subcutaneous injections in the upper arm on Days 1 and 22, followed by 30 min monitoring. On Day 43, mosquito feeding was performed using five uninfected, starved *Ae. aegypti* and *Ae. albopictus* mosquitoes placed on participants’ arms in separate mesh-covered containers for 10–20 min. Importantly, no salivary proteins or mosquito-derived components were injected into participants; all immune responses analysed were elicited by natural mosquito biting post-vaccination. Immunogenicity data, including serological and cellular responses to the vaccine peptides, were collected as part of the original Phase I trial and are reported in detail elsewhere (12). Blood was collected pre-vaccination (day 1), post-vaccination (Day 22) and post-mosquito biting (Day 43) to assess antigen-specific B and T cell responses. T cell responses against whole mosquito saliva were also assessed; however, as participants resided in an *Aedes*-endemic area, they were expected to have experienced prior exposure to mosquito bites.

Redness and swelling at bite sites were assessed 30 min after feeding. Bites and unbitten skin areas were marked with pens to carefully identify the center of each bite site to be biopsied two days later, as well as unbitten skin for the control biopsy. For the biopsy, the area was numbed with 1 to 1.5 mL of lidocaine injected into the dermal layer. The area was disinfected with chloraprep swabs. A total of 3 mm of skin was collected with a punch biopsy tool centered precisely over the bite site where the insect proboscis punctured the skin, using the skin markings as landmarks. Biopsies were immediately immersed in 1.5 mL of RNAlater solution (Invitrogen, Paisley, UK) and stored at +4 °C for shipment to NIAID-NIH for processing. Within 48 h, excess RNAlater solution was removed, and samples were then stored at −80 °C. Samples were later homogenised using porcelain beads and a rotor homogenizer MagNA Lyser (Roche Diagnostics Gmbh, Mannheim, Germany), and RNA was isolated from the RNAlater-preserved skin tissue using the InnuPure-Kit and the InnuPure C16 instrument following manufacturer’s instructions (Analytik Jena, Jena, Germany). RNA yield and quality were determined using a Nanodrop 1000 (Thermo Scientific, Waltham, MA, USA) and the Agilent 2100 Bioanalizer (Agilent Technologies, Santa Clara, CA, USA). Samples with an RIN number above 7 were sent for sequencing. RNA was shipped in dry ice to The Genomic Sciences Laboratory (Room 2518, Thomas Hall, 112 Derieux Place, Raleigh, NC 27695, USA) who prepared cDNA libraries and Illumina RNA sequenced all these samples. Raw data files were sent to the University of Leeds (UK) to conduct the bioinformatic analysis.

### 2.4. Bioinformatic Analyses

Cutadapt software version 5.1 [26] was used to remove low-quality trimmed sequences before the read-paired data was aligned to the human (hg38) genome reference sequences using the splicing-aware STAR aligner with reference to the RefSeq gene annotation. PCR/optical duplicates were detected using Picard tools before the data was sorted by coordinate position and indexed using Samtools software, version 1.13. Transcript read counts were determined using the R package RSubread [27] with and without PCR/optical duplicates. As duplicate reads were comparable, they were included in subsequent analysis. Before statistical analysis, reads data mapping to rRNA transcripts was removed, as was data for low-expressing transcripts (less than three samples had read counts of 5 or more). Differentially expressed transcripts were identified using DeSeq2 [28] with individual variation as a secondary factor. To account for the effects of multiple testing, the statistical significance was adjusted for multiple testing using the Benjamini–Hochberg method as implemented by DeSeq2. Differentially expressed genes were identified as those with an adjusted *p*-value of less than 0.01. Data for the differentially expressed genes was exported to a data frame for further analysis.

The expression profile of each of the samples for the differentially expressed genes was visualised using the R package heatmap to produce a clustered heat map of the log_2_-normalised read count values for the differentially expressed genes. To highlight the differences between samples, colour scaling was performed at the level of each transcript. Heatmap images were cropped to remove redundant sample information. To help identify sample clustering, batch effects, or biological differences, a Principal Component Analysis (PCA) plot of RNA-seq transcriptome analysis was undertaken to visualise variation in gene expression across samples.

Both supervised and unsupervised analyses were undertaken. Supervised approaches use prior knowledge, such as sample labels (e.g., bite vs. control), to classify samples based on known categories. Unsupervised approaches do not rely on predefined labels but instead explore inherent patterns and relationships within the data. This includes our Principal Component Analysis (PCA) and clustering (e.g., hierarchical clustering) which grouped samples based on similarities in gene expression, revealing potential subgroups, batch effects, or novel biological insights without prior assumptions. All data has been uploaded to the publicly available NCBI Bioproject website as PRJNA1279826.

### 2.5. Gene Ontology Analysis

To determine the possible functional effects of the differentially expressed gene on the tissue, a list of all genes present in the experiment was created, along with lists of the differentially expressed genes, upregulated genes, and downregulated genes. In turn, each list of differentially expressed genes was compared to the list of all genes in the experiment using the enrichGO function of the clusterProfiler R package [29]. This function performs a pairwise analysis of the frequency with which gene ontology phrases (in the biological process domain) are linked to genes in each of the gene lists and determines the statistical significance of these changes, which differ between the lists. GO terms were identified as significantly different if the *p*-values were less than 0.05 and q-values (FDR) were less than 0.05 after adjustment for multiple testing using the Benjamini–Hochberg algorithm.

### 2.6. Kegg Pathway Analysis

To identify which KEGG pathways are likely to be affected by the changes in gene expression, the list of the differentially expressed genes was supplied to the enrichKegg function of the clusterProfiler R package, which compared this list to the list of all genes identified in the experiment to determine the statistical probability that each KEGG pathway contained an excess of differentially expressed genes. To identify these pathways, those with a *p*-value less than 0.01 and a q-value of less than 0.05 after adjustment of multiple testing using the Benjamini–Hochberg algorithm were selected, saved to file, and visualised as a dot plot using the dotplot function of clusterProfiler. KEGG pathways that contained a significant number of differentially expressed genes were annotated using the pathview function of the ReactomePA R package [30], such that the genes in question were colour-coded in an image of the pathway.

### 2.7. Immune Deconvolution

Immune deconvolution is a computational method that uses gene expression profiling from bulk RNA sequencing data to estimate the relative proportion of different immune cell types present within tissue samples. Utilising reference expression profiles and expression markers, deconvolution markers infer immune cell composition [31], offering an alternative to methods that require single-cell solutions, like flow cytometry. Gene transcript counts generated from bulk RNA sequencing of *Ae. aegypti*- or *Ae. albopictus*-bitten or unbitten skin following vaccination with either placebo or different formulations of the AGS-v PLUS vaccine were generated and normalised to transcripts per kilobase million (TPKM). TPKM normalised gene counts were summed together in genes containing multiple transcripts using version 4.3.3 of R and version 1.4.2206 of R studio. Immune deconvolution was then performed on normalised gene transcript counts using the TIMER2.0 online application (http://timer.cistrome.org/) [32]. From the available algorithms, QuantiSeq [33] was selected as the most appropriate output due to its ability to accurately quantify a broad spectrum of immune populations.

### 2.8. Gene Nomenclature

All genes are referred to using NCBI reference gene names. For a full list of gene abbreviations, see Appendix A.

## 3. Results

A total of 51 participants (randomised 1:1:1:1:1) received one of five dosing regimens: (i) two doses of saline placebo, (ii) two doses of non-adjuvanted AGS-v PLUS, (iii) one dose of Montanide ISA-51-adjuvanted AGS-v PLUS followed by saline placebo, (iv) two doses of Montanide ISA-51-adjuvanted AGS-v PLUS, or (v) two doses of Alhydrogel-adjuvanted AGS-v PLUS. Twenty-one days after the final injection, all participants were subjected to bites from either *Ae. aegypti* or *Ae. albopictus* mosquitoes at separate sites. At 48 h post-bite, skin biopsies were taken from either resting unbitten skin, *Ae. aegypti*-bitten skin, or *Ae. albopictus*-bitten skin. RNA was extracted and subjected to RNA-seq analysis to define differentially expressed genes (DEGs).

### 3.1. Mosquito Biting Resulted in Large Numbers of DEGs by 48 h

To characterise the unmodified cutaneous immune response to *Aedes* mosquito biting, we first analysed data from placebo-vaccinated participants. This provided a reference against which to interpret the effects of AGS-v PLUS vaccination. We therefore restricted our initial analysis to participants who had received only placebo saline vaccination prior to mosquito exposure. Following an initial examination of the dataset, several thousand DEGs were observed in bitten skin. To enhance stringency, the threshold for DEG identification was set at *p* < 0.01. This filtering approach ensured that only the most statistically robust changes were considered. Total numbers of DEGs were similar irrespective of gender or race of the participant. Instead, individual variability was the largest source of variation, most likely reflecting differences in prior and current exposure to mosquito bites. Accordingly, DEGs were identified by controlling for individual variation. Biting by either of the *Aedes* species resulted in widespread upregulation of genes (Figure 1). Irrespective of mosquito species, the overall induction of DEGs in bitten skin remained distinct from resting, unbitten skin, as assessed by heatmap or by Principal Component Analysis (PCA) plots. These findings suggest a conserved human immune response to *Aedes* mosquito bites, independent of mosquito species-specific differences.

Transcriptome analyses identify the global immune signature of *Aedes* mosquito-bitten human skin.

We conducted KEGG pathway and gene ontology analyses to identify functional gene modules upregulated by mosquito biting, focussing on *Ae. aegypti* biting in the first instance, using an unsupervised approach. This approach does not rely on predefined labels but instead explores inherent patterns and relationships within the data. Importantly, we found that most upregulated gene modules were immune/inflammatory in nature, with high numbers of significantly upregulated genes in each array/module (Appendix A). The most enriched pathways included those associated with innate immunity, chemokine activity, TNF binding activity, cytokine receptor activity, interferon (IFN) responses, and NF-κB signalling.

Adaptive immune response activation was also evident, with significant upregulation of pathways related to antigen processing and presentation, T cell function, and B cell biology. Interestingly, given the known vascular permeability changes associated with mosquito bites in mice [16,18], human bitten skin exhibited a strong transcriptional signature indicative of endothelial cell activation. This included upregulation of genes involved in leukocyte trans-endothelial migration, cell–cell adhesion, and vascular remodelling.

To highlight individual gene units within these upregulated pathways, KEGG plots were generated in which significantly upregulated genes are marked in red (Appendix A). There was widespread and robust activation of immune/inflammatory pathway gene units including cytokine–cytokine receptor interaction gene units, leukocyte trans-endothelial migration, chemokine gene units, T cell receptor signalling, antigen processing and presentation, RAS signalling, MAPK signalling, IL-17 and Th17 T cell differentiation, RAP1 signalling, B cell receptor signalling, Th1 and Th2 cell differentiation, TNF signalling, and VEGF signalling (Appendix A). Also dysregulated were cell junction pathways including tight junctions, focal adhesion, GAP junctions, adherens, and regulation of actin cytoskeleton (Appendix A), and innate immune signalling pathways including RIG-I like receptor, NOD-like receptor Toll-like receptor (TLR), and NK-κB signalling pathways (Appendix A).

### 3.2. Mosquito Biting Results in Upregulation of Both Innate Immune and Adaptive Immune Genes

To further dissect the immune response of *Ae. aegypti* and additionally compare this to *Ae. albopictus* mosquito-bitten skin, we performed a supervised analysis focusing on immune/inflammatory genes expressed by mosquito-bitten skin that are implicated in host susceptibility to virus [3,17,25] (Figure 2, Appendix A). Here, we generated a list of the 400 most upregulated genes (*p* > 0.01) in *Ae. aegypti*-bitten skin, clustered them based on known immune/inflammatory function, and plotted the gene expression fold change alongside the corresponding fold change in *Ae. albopictus*-bitten skin. Among the most upregulated genes following both *Ae. aegypti* and *Ae. albopictus* bites were chemokines, with prominent upregulation including CXCL9, CXCL10, and CXCL11 (which bind to CXCR3, and recruit activated CD8 and Th1 CD4 T cells), as well as CCL5 and CCL3 that similarly recruit these leukocytes. Markers of Th1 CD4 and CD8 T cell activation were also highly elevated, including T-bet (Th1 master regulator), CXCR3, CD4, CD8, IL-12RB, and granzyme B, amongst others. In contrast, Th2 master genes such as GATA3, IL-4, and IL-5 were not upregulated, although a modest increase in Th2-associated IL-13 and CCR4 gene expression was observed. Th17 master regulator ROR-γT and IL-17 were also absent. These results suggest a predominant Th1-skewed immune response in human skin following mosquito bites, with potential implications for antiviral immunity.

Further highly upregulated genes included CSF2/3 and LTB, indicative of myeloid cell activation and tissue remodelling consistent with wound repair. Consistent with previous mouse studies [19], genes associated with epidermal inflammation (e.g., IL-6, IL1B, S100A7, S100A8, and S100A9) and matrix metalloproteases (MMPs) were also significantly upregulated, as were E- and L-selectin, which are required for leukocyte entry into skin. Several pattern recognition receptors and IFN-simulated genes were also upregulated including FPR2, members of the CLEC family, and—notably—APOBEC3A.

Interestingly, while the overall immune response was similar between the two mosquito species, *Ae. albopictus* bites led to a stronger induction of several genes, including chemokines (e.g., CXCL9/10/11, CCL3, CCL18, CL22, CCL5), inflammatory mediators (e.g., MMP1, S100A7-9), and pattern recognition receptors/interferon-stimulated genes (e.g., FRP2, TLR8, APOBEC31, CLEC4C). Unique to *Ae. albopictus*-bitten skin was the upregulation of CCL7, CCL20, CXCL5, IL-27, and IFN-γ. In contrast, IL-6 was one of the few genes more highly upregulated following *Ae. aegypti* bites.

### 3.3. Immune Deconvolution Identifies Significant Increases in Leukocyte Signatures

Given the robust upregulation of chemokines and MMPs, we decided to utilise immune deconvolution to infer immune cell recruitment in *Ae. aegypti* and *Ae. albopictus*-bitten skin (Figure 3). Neutrophil and monocyte signatures with mosquito biting were mostly unchanged or absent. This is consistent with mouse models in which myeloid cell infiltrates peak within minutes to hours but then decrease by 48 h [18,19,24], which helps validate the use of this deconvolution approach in studying mosquito-bitten skin. In contrast to the absence of neutrophils, we observed significant increases in CD4, CD8, NK cells, dendritic cells and macrophages—all cell types that are recruited by the upregulated chemokines identified above.

### 3.4. Vaccination with AGS-v PLUS Modulates the Skin’s Response to Mosquito Biting

Next, we assessed whether prior vaccination with AGS-v PLUS modulated the gene expression response to biting as identified above. Here, participants vaccinated with AGS-v PLUS also exhibited widespread transcriptomic changes in response to either *Ae. aegypti* mosquito bites (Figure 4) or *Ae. albopictus* bites (Figure 5). PCA plots showed distinct clustering of bitten versus unbitten skin, regardless of vaccination status. However, individuals who received two doses of AGS-v PLUS with adjuvant (either Alhydrogel or Montanide ISA-51) exhibited broader transcriptomic shifts compared to those vaccinated without adjuvant. This was most clearly depicted by the increased number of genes with a −Log10*p* value > 50 on the volcano plots (Figure 4Di and Figure 5Di). There was also a shift in heat map clustering and increased numbers of upregulated DEG in *Ae albopictus*-bitten skin for adjuvanted AGS-v PLUS individuals (Figure 5Biii–Diii) compared to those receiving non adjuvanted AGS-v PLUS (Figure 5Aiii). These findings indicate that adjuvanted AGS-v PLUS may enhance the immune response to the peptides in the vaccine and therefore have a more pronounced effect against whole mosquito saliva.

To better assess whether prior vaccination with adjuvanted AGS-v PLUS modulated expression of the specific immune/inflammatory gene modules identified above (Figure 2), we compared fold change in response to biting in participants that had received placebo to those that had received AGS-v-PLUS (Figure 6). Importantly, those individuals that received a prime/boost administration of Montanide-adjuvanted AGS-v PLUS exhibited higher fold induction of many key immune genes that have known antiviral function upon biting with either *Ae. aegypti* or *albopictus*. This included several T cell-attracting chemokines including the CXCR3 ligands CXCL9/10/11. Correspondingly, several key Th1 CD4 and CD8 genes were more potently upregulated including granzyme B, IFN-γ, and T-bet. As for individuals that received placebo, Th2- and Th17-associated genes were generally not upregulated by mosquito biting. Monocyte attractants CCL7 and CCL8 were also more potently upregulated in Montanide AGS-v PLUS-vaccinated individuals. Importantly, several of the genes in the Pattern Recognition Receptor and IFN-stimulated module were more potently upregulated by biting in those individuals that received a prime/boost administration of Montanide-adjuvanted AGS-v PLUS. These genes included TLR7, TLR8, FPR2, CLEC4C, CLEC4D, CLEC4E, ZBP1, APOBEC3A, RSAD2, ZBP1, ISG15, MX1, IFH1 and OAS1A. Many of these genes have well-characterised potent innate immune antiviral functions. Those participants that received only one dose of AGS-v PLUS in Montanide ISA-51 or two doses of non-adjuvanted AGS-v PLUS did not exhibit such widespread increases in immune gene expression. Notably, those individuals receiving a prime/boost of AGSv PLUS with the Alhydrogel adjuvant exhibited suppressed upregulation of these genes in response to biting, compared to those receiving placebo, suggesting this vaccination formulation tolerised inflammatory response to biting. Together, this shows that type of adjuvant and number of vaccine doses administered are key in optimising vaccine effectiveness.

Interestingly, not all immune genes were more strongly upregulated in participants receiving a primer/boost of Montanide-adjuvanted AGS-v PLUS vaccination. Instead, a small number of key inflammatory genes had lower expression compared to bitten participants that received placebo alone. This included a crucial set of genes that have been implicated in increased susceptibility to arbovirus infection [19], including neutrophil-attracting chemokines (CXCL1, CXCL3, CXCL8). Correspondingly, neutrophil-specific CXCR2 expression (Figure 6, inflammatory mediator panel) was absent in AGS-v PLUS-vaccinated participant skin, and prototypic markers of neutrophil-associated skin inflammation (S100A7, S100A8, S100A9, CSF2) were also reduced. These gene expression differences are most apparent when directly comparing gene fold change between participants receiving placebo and a primer/boost administration of Montanide-adjuvanted AGS-v PLUS receiving participants (Appendix A).

Finally, immune deconvolution revealed no significant change in most leukocyte subset signature abundance with vaccination status (Figure 7). However, there were significant increases in the skin signature of recipients that received two doses of AGS-v PLUS with Montanide ISA-51 for NK cells following *Ae. aegypti* bites, and separately for dendritic cells following *Ae. albopictus* bites (alongside a non-significant trend (*p* = 0.057) of increased DC in *Ae. aegypti* bites) (Figure 7). Together, these findings suggest that a primer/boost administration of AGS-v PLUS in Montanide ISA-51 may more robustly fine-tune the immune response to mosquito bites, potentially enhancing antiviral defences.

## 4. Discussion

This study provides a comprehensive gene expression analysis of the response of human skin to *Aedes* mosquito bites and evaluates how vaccination with AGS-v PLUS can modulate this response. Our findings highlight that the natural human response to the bite of female *Aedes* species mosquitoes is characterised by a predominant Th1-skewed signature with evidence of endothelial activation and chemokine expression. Importantly, we demonstrate that AGS-v PLUS vaccination in the right dosage and formulation can modulate this response, enhancing some antiviral pathways while attenuating some undesirable pro-inflammatory signatures. These findings have potential implications for mosquito-borne disease susceptibility and vaccine development.

Mosquito saliva has evolved to facilitate efficient blood feeding of vertebrates. As such, it contains myriad factors that have biologically potent effects on vertebrate skin, including factors that inhibit blood clotting, enhance vascular permeability, and modulate immune responses. It is into this unique tissue microenvironment that the virus is deposited during mosquito probing of the skin. Crucially, the skin’s response elicited by saliva enhances infection of host cells by multiple genetically distinct viruses [5,17,34]. Therefore, targeting mosquito salivary proteins to modulate these host responses through vaccination could be an effective strategy, as compared to targeting viral antigens alone. Supporting this overall concept, studies in animal models have demonstrated that immunization with sand fly salivary antigens can help prevent leishmania infection [5,35].

In our study, we found that the human immune response to mosquito biting was largely conserved between *Ae. aegypti* and *Ae. albopictus*, with both species inducing a pronounced inflammatory reaction. While some differences in gene expression were observed, these may likely reflect differences in prior exposure to mosquito saliva rather than fundamental biological differences between the species. *Ae. albopictus* is well established in Baltimore, where the study participants were located, whereas *Ae. aegypti* is not [36]. Notably, our findings align with previous studies in mice, where mosquito saliva has been shown to modulate immune responses in a manner that can influence viral infection outcomes [3,16,17,19,24,37]. The observed upregulation of pathways related to vascular permeability, endothelial cell activation, and leukocyte migration suggests that mosquito bites induce a complex local immune response that extends beyond immediate inflammation and may influence tissue remodelling and immune cell infiltration. Notably, the upregulation of TLR4-related genes aligns with previous findings that *Aedes* mosquito saliva contains a TLR4-activating factor capable of modulating host susceptibility to e.g., Zika virus infection in mice [23,38]. Many of the immune responses elicited by *Aedes* mosquito biting activate antiviral pathways, such as IFN-γ, which raises the question as to why these do not inhibit virus during mosquito-borne virus infection. However, for such immune responses to be sufficiently effective at inhibiting virus, they must be efficiently and robustly activated in a timely manner [3]. In mouse models, virus replicates and disseminates to blood within 24–48 h [19,22,39]. Therefore, for cutaneous antiviral immune responses to be effective, they must be rapid and robust.

We also assessed how prior AGS-v PLUS vaccination modulated the response to mosquito biting taking place three weeks post-vaccination. This sequential comparison enabled us to distinguish transcriptomic changes due to biting alone from those modified by vaccination. Primer/booster vaccination with Montanide ISA-51-adjuvanted AGS-v PLUS enhanced multiple antiviral pathways and primed the skin for a stronger immune reaction. Notably, NK and dendritic cell signatures increased with adjuvanted vaccination, while neutrophil-associated genes were reduced; this is a potentially beneficial shift, as neutrophil-driven inflammation can enhance viral susceptibility in mouse models [8,19,23]. While some leukocyte populations (e.g., NK cells, dendritic cells) were more abundant post-bite in vaccinees, this recruitment occurred alongside increased expression of IFN-stimulated genes, cytotoxic effectors (e.g., granzyme B), and Th1-associated transcription factors like T-bet. This suggests these cells were likely activated and antiviral, rather than permissive to infection. In contrast, monocyte signatures, which may mark susceptible targets in mouse skin, were not significantly elevated. These findings underscore that immune cell recruitment alone does not predict susceptibility; the phenotype and activation state of cells are also critical. Although most leukocyte subset signatures were unchanged at 48 h post-bite, vaccine-induced increases in key antiviral cell types could contribute to improved control. It is also important to note that our analysis focused on a single timepoint, which was selected to capture peak gene expression. Other subsets, particularly early innate responders, may be differentially recruited at earlier stages. Future studies should explore how AGS-v PLUS shapes immune responses over time.

While this study provides a key transcriptomic view of the human skin immune response to mosquito biting, including modulation by prior vaccination with AGS-v PLUS, there are nonetheless limitations with this approach. First, although the cohort size was appropriate for a Phase 1 trial, the number of participants per treatment arm limited our ability to explore how demographic variables, such as age, gender, or ethnicity, may influence immune responses. A larger, more powered study would be necessary to detect subtle subgroup differences. Second, while transcriptomic profiling provides a valuable global picture of gene expression, it cannot definitively identify the specific cell types responsible for gene expression changes or their activation states. Our use of immune deconvolution methods helps infer likely cell populations but lacks the resolution of techniques such as flow cytometry or single-cell RNA sequencing. Future studies should incorporate direct immune phenotyping of skin biopsies, including flow cytometric analysis of infiltrating leukocytes, to validate and expand upon these findings. Third, the focus on the 48 h timepoint, which was selected to match peak inflammatory signatures, limits insight into earlier or later events in the response. A time-course study assessing skin responses at multiple intervals post-bite would better define the kinetics of leukocyte recruitment and inflammatory resolution. Fourth, while the primary focus of this study was on the effect of AGS-v PLUS on bitten skin, there are alternative or complementary angles that were not fully explored. These include comparing systemic immune responses in blood to local responses in skin and performing paired proteomic or cytokine analyses. These limitations underscore the importance of integrating multimodal immunological approaches and larger cohorts in future trials to fully define the mechanisms by which such vaccines might confer protection against arboviral infection. Finally, no baseline seroreactivity to whole mosquito saliva was assessed. This decision was made as participants were recruited from a region with established *Aedes* mosquito populations and thus likely had prior exposure to mosquito bites. Using placebo-vaccinated individuals as comparators served as a more practical and ecologically valid control. Pre- and post-vaccination immune responses specific to the vaccine peptides themselves were captured as part of the original trial and are detailed in a separate publication [12].

In summary, these findings highlight the potential of vector-targeted vaccines to shape host immune responses at the site of pathogen transmission. By grouping gene expression data into functionally annotated immune signatures, we distilled the extensive dataset into interpretable and immunologically relevant patterns. We suggest that vaccines like AGS-v PLUS could modulate the immune environment of the skin, providing an additional layer of protection against multiple genetically distinct mosquito-borne diseases. Future studies should explore the long-term effects of vaccination as well as the functional consequences of the observed immune modulation for viral susceptibility and disease outcomes. In conclusion, this study provides new insights into the immunological impact of mosquito bites on humans and the potential for vaccination to modify these responses. Understanding how mosquito saliva interacts with the host immune system is critical for developing new strategies to combat vector-borne diseases, and these findings suggest that targeting the mosquito–host interface could be a promising avenue for future interventions.

## Figures and Tables

**Figure 1 vaccines-13-01026-f001:**
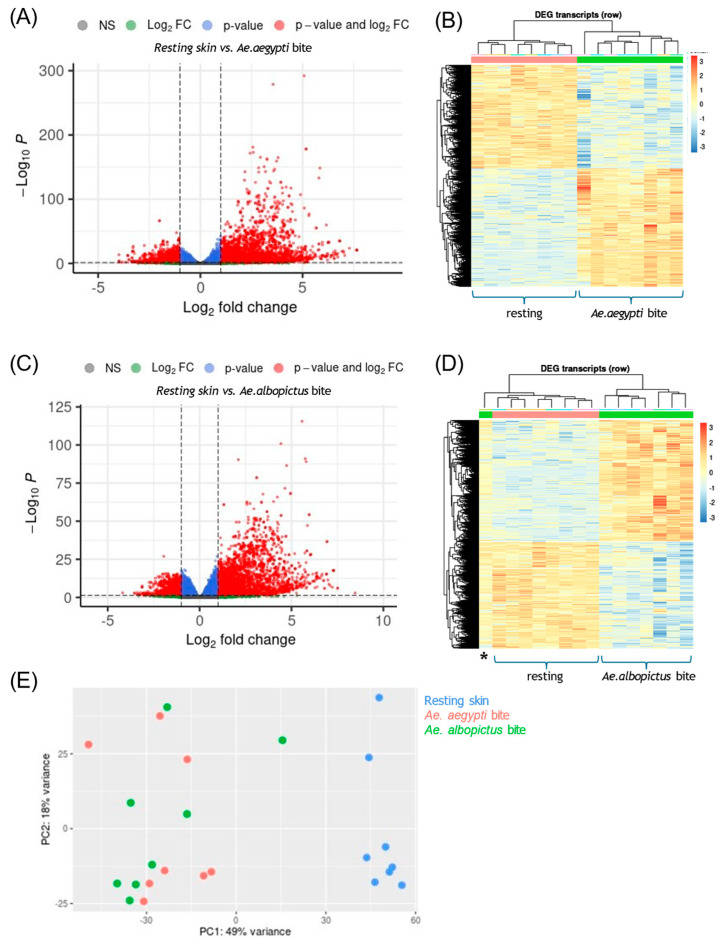
*Aedes* genus mosquito biting results in large numbers of DEGs in human skin. Participants that received placebo saline control were subject to mosquito biting by either *Ae. aegypti* (**A**,**B**) or *Ae. albopictus* (**C**,**D**) mosquitoes at discreet sites remote from prior placebo saline injection (n = 8). At 48 h post-bite, skin was biopsied, RNA extracted, and RNA-seq performed. Differential expression was computed with a paired design controlling for individual (DESeq2, Benjamini–Hochberg FDR); unless stated otherwise, significance reflects adjusted *p* < 0.01. Fold change is reported relative to each participant’s unbitten skin. (**A**,**C**) Volcano plots for *Ae. aegypti* (**A**) and *Ae. albopictus* (**C**), highlighting genes upregulated ≥2-fold (green), significant at FDR < 0.01 (blue), or both (red). (**B**,**D**) Heat maps (log2 scale) for *Ae. aegypti* (**B**) and *Ae. albopictus* (**D**) bitten skin, showing global expression patterns; rows are transcripts (row-scaled), columns are samples, hierarchically clustered. Samples cluster by condition (unbitten vs. bitten). The sample marked * in (**D**) is an *albopictus*-bitten biopsy that clustered with unbitten controls. (**E**) Principal Component Analysis showing clear separation of unbitten (blue), *Ae. aegypti*-bitten (red), and *Ae. albopictus*-bitten (green) samples. Together, these analyses show that *Aedes* biting elicits extensive transcriptional reprogramming at 48 h.

**Figure 2 vaccines-13-01026-f002:**
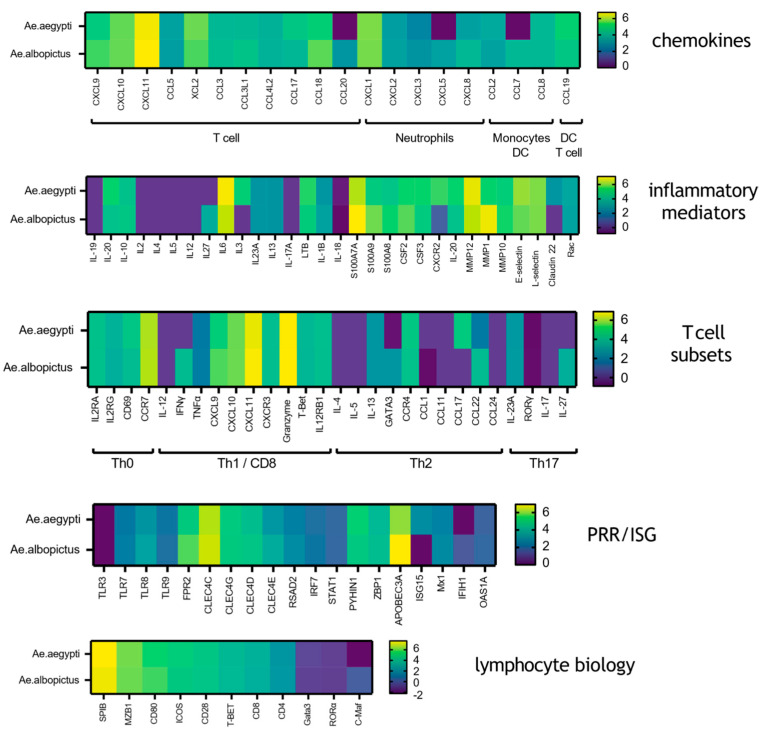
Mosquito biting results in upregulation of both innate immune and adaptive immune genes. Immune/inflammatory gene expression was defined by an unsupervised analysis of skin biopsies from placebo-vaccinated participants 48 h after *Ae. aegypti* and *Ae. albopictus* mosquito biting. Those DEG that were most highly upregulated were then selected and clustered into functional groupings for chemokines, inflammatory mediators, T cell subsets, pattern recognition receptors (PPRs) and interferon stimulated genes (ISGs) and lymphocyte biology. Heat maps depicting fold change on a log_2_ scale compared to unbitten skin are shown. Genes associated with specific cellular subsets are indicated within the figure.

**Figure 3 vaccines-13-01026-f003:**
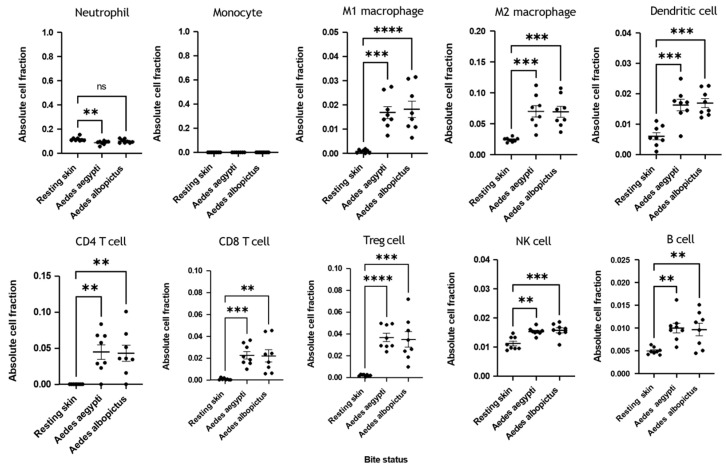
Immune deconvolution reveals modulated leukocyte signature at 48 h post-mosquito bite. Participants that received placebo saline control were subject to biting by mosquitoes at discreet sites remote from prior placebo saline injection. At 48 h post-bite, skin was biopsied, RNA extracted, and RNA-seq performed. Immune deconvolution of the transcriptome data was used to estimate the proportion of different immune cell types present at the biopsy. Transcript counts were normalised for each condition to transcripts per kilobase million, with duplicated transcript reads within each group summed using R. Absolute cell fractions were estimated using the TIMER2.0 online immune deconvolution tool. A one-way ANOVA with Dunnett’s post-test was performed—** *p* < 0.01, *** *p* < 0.001, **** *p* < 0.0001, ns = non-significant.

**Figure 4 vaccines-13-01026-f004:**
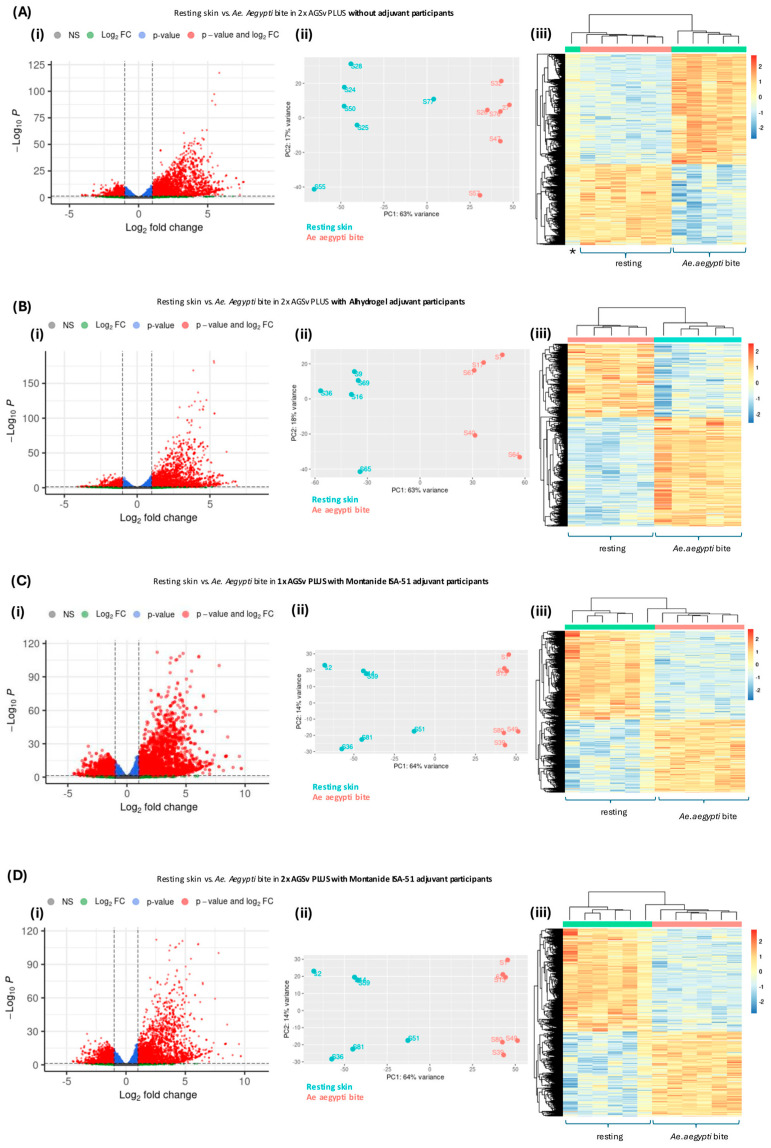
Transcriptomic response to *Ae. aegypti* biting in AGS-v PLUS vaccinees. Healthy volunteers received one of four AGS-v PLUS regimens: (**A**) two doses without adjuvant, (**B**) two doses with Alhydrogel adjuvant, (**C**) a single dose with Montanide ISA-51, or (**D**) two doses with Montanide ISA-51. At 48 h post-bite, participants were subjected to controlled *Ae. aegypti* mosquito feeding at sites remote from prior vaccine injections. Matched unbitten skin biopsies from the same participants served as controls. Total RNA was extracted, sequenced, and analysed by DESeq2 with participant as a paired factor; adjusted *p* < 0.01 (Benjamini–Hochberg) was used to define differentially expressed genes (DEGs). For each regimen: (**i**) Volcano plots highlight genes upregulated ≥2-fold (green), significant at FDR < 0.01 (blue), or both (red). (**ii**) Principal Component Analysis (PCA) shows separation of bitten versus unbitten samples. (**iii**) Heat maps display log2 fold changes across DEGs, with samples clustering primarily by condition. The sample marked with an * in (**Aiii**) corresponds to a mosquito-bitten biopsy that grouped with unbitten controls. Overall, AGS-v PLUS vaccination modified the transcriptional response to *Ae. aegypti* biting.

**Figure 5 vaccines-13-01026-f005:**
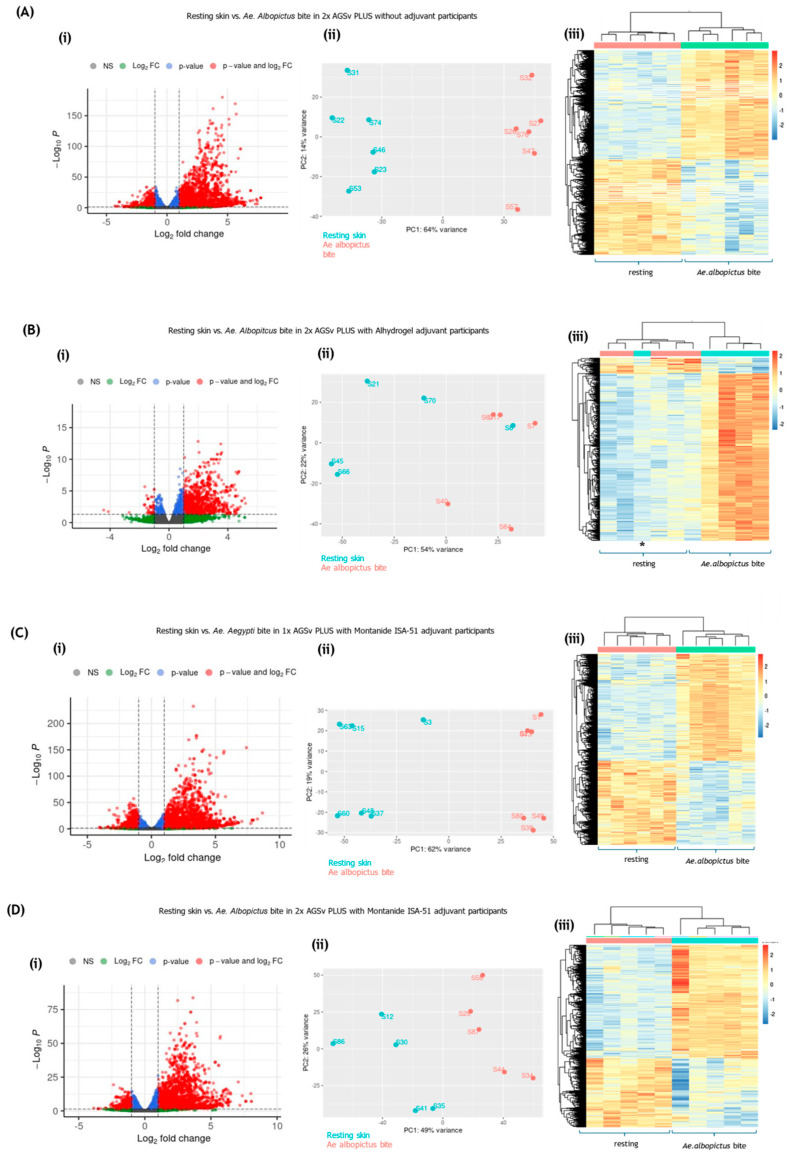
Transcriptomic response to *Ae. albopictus* biting in AGS-v PLUS vaccinees. Healthy volunteers received one of four AGS-v PLUS regimens: (**A**) two doses without adjuvant, (**B**) two doses with Alhydrogel adjuvant, (**C**) a single dose with Montanide ISA-51, or (**D**) two doses with Montanide ISA-51. At 48 h post-bite, participants were subjected to controlled *Ae. albopictus* mosquito feeding at discrete sites remote from vaccination. Matched unbitten skin biopsies from the same participants were used as controls. Total RNA was extracted, sequenced, and analysed by DESeq2 with participant as a paired factor; adjusted *p* < 0.01 (Benjamini–Hochberg) was used to define differentially expressed genes (DEGs). For each regimen: (**i**) Volcano plots highlight genes upregulated ≥2-fold (green), significant at FDR < 0.01 (blue), or both (red). (**ii**) Principal Component Analysis (PCA) shows clustering of bitten versus unbitten skin. (**iii**) Heat maps display log2 fold changes across DEGs, with samples clustering primarily by condition. The sample marked with an * in (**Biii**) corresponds to a mosquito-bitten biopsy that grouped with unbitten controls. Overall, AGS-v PLUS vaccination altered the magnitude and pattern of skin gene expression after *Ae. albopictus* bites.

**Figure 6 vaccines-13-01026-f006:**
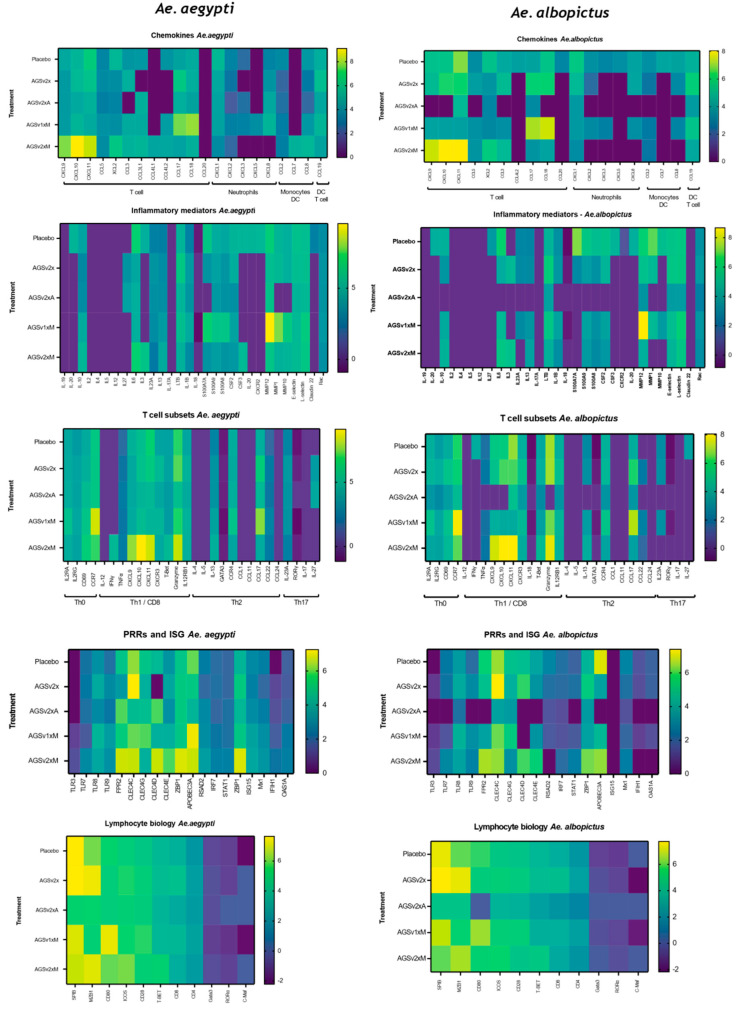
AGSv-PLUS vaccination modulates expression of antiviral transcripts in response to mosquito biting. Immune/inflammatory gene expression was defined in skin biopsies from vaccinated participants 48 h after *Ae. aegypti* and *Ae. albopictus* mosquito biting, compared to unbitten skin. Participants were administered with either saline placebo or AGS-v PLUS with or without adjuvant (A, Alhydrogel; M, Montanide), either prime (1×) or prime/boost, 22 days apart (2×), and then subjected to biting by mosquitoes at discreet sites remote from prior injection. Those DEGs that were most highly upregulated compared to resting unbitten skin were then selected and clustered into functional groupings for chemokines, inflammatory mediators, T cell subsets, pattern recognition receptors (PPRs), interferon stimulated genes (ISGs), and lymphocyte biology. Heat maps depicting fold change on a log_2_ scale compared to unbitten skin are shown. For chemokines, those gene products that are associated with recruitment of specific cellular subsets are indicated within the figure. For T cell subsets, genes associated with specific Th and CD8 lineage are shown.

**Figure 7 vaccines-13-01026-f007:**
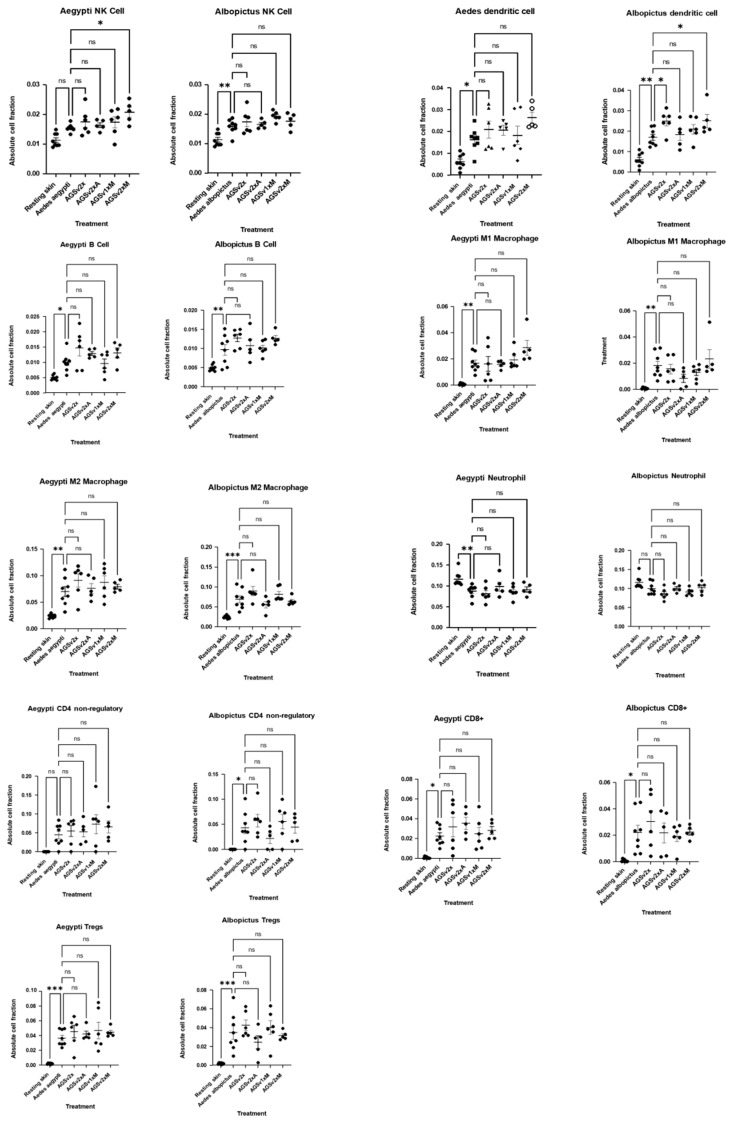
Immune deconvolution identifies a signature of increased numbers of some leukocyte in vaccinated individuals 48 h post-mosquito biting. Participants were administered with either saline placebo or AGS-v PLUS with or without adjuvant (A, Alhydrogel; M, Montanide), either prime (1×) or prime/boost, 22 days apart (2×); they were then subjected to biting by either *Ae. aegypti* or *Ae. albopictus* mosquitoes at discreet sites remote from prior injection, except for resting skin which was not mosquito-bitten. Data derived from the skin biopsy sites of all these participants were assessed for leukocyte signatures using TIMER2.0. Here, immune deconvolution was conducted by normalizing transcript counts for each condition to transcripts per kilobase million, with duplicated transcript reads within each group summed using R. A one-way ANOVA with Dunnett’s post-test was performed—* *p* < 0.05, ** *p* < 0.01, *** *p* < 0.001, ns = non-significant.

## Data Availability

All data has been upload to the publicly available NCBI Bioproject website, as PRJNA1279826.

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
