# Peer review of "AGS-v PLUS, a Mosquito Salivary Peptide Vaccine, Modulates the Response to Aedes Mosquito Bites in Humans"

_vaccines, 2025, doi:10.3390/vaccines13101026_

Round 1
Reviewer 1 Report
Comments and Suggestions for Authors
The study by Barningham et al. demonstrated an immune response signature based on the transcriptome of human skin from volunteers after contact via subcutaneous inoculation of Aedes mosquito saliva, followed or not by subcutaneous inoculation of different combinations of the AGS-v PLUS vaccine. Although the number of human skin samples was limited, the author employed an expensive methodology that allowed the identification of many genes, resulting in a substantial amount of data. The author emphasized that the study's limitations included the limited number of specimens. Due to the enormous amount of data generated, the main conclusion described in lines 528-530 of the discussion is challenging to visualize. Nevertheless, the authors generated extensive and interesting data, which will undoubtedly contribute to this underdeveloped area of research. Some comments to improve the description of the study:
- The abstract should clearly state the objective, methodology, and results of the study, providing greater clarity to the reader about the research's purpose and findings.
- The authors should provide a brief explanation of the AGS-v PLUS vaccine and its components, particularly the five standard mosquito saliva peptides, to give the reader a better understanding of the vaccine used in the study.
- The authors could modify the scales of absolute fractions of neutrophil cells and monocytes in Figure 3 for better visualization of the data.
- How do the authors discuss the specificity of the response found in human skin based on the signature of the immune response to Aedes mosquito saliva and the AGS-v PLUS vaccine?
- Could the authors discuss what they would expect or have additional data that would demonstrate the immune response signature in response to DENV-infected Aedes mosquito saliva? Also, have the study participants ever had dengue fever?
- How do the authors discuss the specificity of the response found in human skin based on the signature of the immune response to Aedes mosquito saliva and the AGS-v PLUS vaccine?
- Could the authors discuss what the immune response signature would be in response to the AGS-v PLUS vaccine, followed by the addition of DENV-infected Aedes mosquito saliva to assess vaccine protection? Do the authors envision additional experiments evaluating the protective antiviral role of the AGS-v PLUS vaccine?
Author Response
All responses by authors are indicated with a “>>” and are in red font.
All changes to manuscript are indicated with use of red font to help guide assessment.
Review 1
Comments and Suggestions for Authors
The study by Barningham et al. demonstrated an immune response signature based on the transcriptome of human skin from volunteers after contact via subcutaneous inoculation of Aedes mosquito saliva, followed or not by subcutaneous inoculation of different combinations of the AGS-v PLUS vaccine. Although the number of human skin samples was limited, the author employed an expensive methodology that allowed the identification of many genes, resulting in a substantial amount of data. The author emphasized that the study's limitations included the limited number of specimens. Due to the enormous amount of data generated, the main conclusion described in lines 528-530 of the discussion is challenging to visualize. Nevertheless, the authors generated extensive and interesting data, which will undoubtedly contribute to this underdeveloped area of research.
>> We thank the reviewer for their time and expertise in reviewing our manuscript. We believe that addressing these points has strengthened the study, improved its clarity, and enhanced its potential impact.
Some comments to improve the description of the study:
The abstract should clearly state the objective, methodology, and results of the study, providing greater clarity to the reader about the research's purpose and findings.
>> We agree and have revised the abstract to more clearly articulate the study’s objective, methodology, and main findings. We now explicitly state that participants were first vaccinated and subsequently exposed to mosquito bites. To clarify a key point of confusion, we note that no salivary antigens were injected at any stage; the immune responses studied were solely those elicited by mosquito biting three weeks post-vaccination. We have also updated the introduction to reinforce this clarification and improve overall clarity.
>> We agree that this study generated a large amount of data. By grouping gene expression data into functionally annotated immune signatures, we distilled the extensive dataset into interpretable and immunologically relevant patterns. We hope that the re-worded text better clarifies this.
The authors should provide a brief explanation of the AGS-v PLUS vaccine and its components, particularly the five standard mosquito saliva peptides, to give the reader a better understanding of the vaccine used in the study.
>> Thank you for this helpful suggestion. We have now included a brief explanation of AGS-v PLUS in the Introduction, including the nature of the five synthetic mosquito salivary peptides used in the formulation. This additional context should aid readers in understanding the vaccine’s design and intended mechanism of action.
The authors could modify the scales of absolute fractions of neutrophil cells and monocytes in Figure 3 for better visualization of the data.
>> Thank you for this observation. We have adjusted the figure legends to better reflect the visualization of neutrophil data. As monocyte values were consistently near zero across groups, we have clarified this explicitly in the main text, which should help readers interpret these values appropriately.
How do the authors discuss the specificity of the response found in human skin based on the signature of the immune response to Aedes mosquito saliva and the AGS-v PLUS vaccine?
>> We appreciate that this point may not have been sufficiently clear in the original manuscript. To address this, we have revised the relevant sections throughout the manuscript to clarify that we first analysed the gene expression response to mosquito biting in placebo-vaccinated participants, to define the natural human skin immune response to mosquito saliva—an area that remains poorly characterised. We then assessed how these responses were modulated in participants who received AGS-v PLUS. This sequential approach allowed us to define both the unmodified and vaccine-altered immune response signatures to mosquito bites. This approach enables direct comparison within the same biological context (i.e., human skin), controlling for individual variability.
Could the authors discuss what they would expect or have additional data that would demonstrate the immune response signature in response to DENV-infected Aedes mosquito saliva? Also, have the study participants ever had dengue fever?
>> As the study was conducted in a non-endemic region (Baltimore, USA), it is unlikely that participants had prior dengue virus (DENV) infection. We did not screen participants for DENV serostatus. While defining the skin immune response to bites from DENV-infected mosquitoes would indeed provide key insights into infection biology, this falls beyond the scope of the current study, which focused on host responses to mosquito biting in the absence of viral infection, and on how these responses are modulated by vaccination. We agree that this is a highly valuable future direction.
How do the authors discuss the specificity of the response found in human skin based on the signature of the immune response to Aedes mosquito saliva and the AGS-v PLUS vaccine?
>> As addressed above, we clarified that our study design enabled us to first define the baseline immune transcriptomic signature in response to mosquito biting (using placebo recipients) and then evaluate how this is modulated by prior vaccination with AGS-v PLUS. We have revised the relevant text throughout the manuscript to more clearly explain this two-step analysis and how it allowed us to identify vaccine-specific modulation of the skin immune response to mosquito saliva.
Could the authors discuss what the immune response signature would be in response to the AGS-v PLUS vaccine, followed by the addition of DENV-infected Aedes mosquito saliva to assess vaccine protection? Do the authors envision additional experiments evaluating the protective antiviral role of the AGS-v PLUS vaccine?
>> We appreciate this thoughtful suggestion. A study evaluating the skin immune response to DENV-infected mosquito bites following AGS-v PLUS vaccination would be highly informative. However, such a study would currently be ethically and logistically challenging in humans, as intentional exposure to DENV via mosquito bite is not permitted. A logical next step would be to investigate this question in animal models, such as mice engineered to mimic human skin immune responses. While these experiments fall outside the scope of the current study, we agree that this is a key future direction to establish the functional antiviral efficacy of AGS-v PLUS in the context of live virus challenge.
Reviewer 2 Report
Comments and Suggestions for Authors
The following manuscript entitled “Ags-V Plus, a Mosquito Salivary Peptide Vaccine, Modulates the Human Response to Aedes Mosquito Bites” by Barningham and colleagues submitted to Vaccines, the authors assessed human immune responses to a novel vaccine, AGS-v PLUS, containing antigens derived from mosquito salivary proteins. Vaccines and placebo-injected groups were included in the study and the cutaneous response to mosquito biting was used as primary outcome. Overall, the trial identified that the adaptive immune response markers such as CD4 Th1 and CD8 T cells signatures and leukocyte influx at 48 hours post-bite suffered significant changes in vaccinees vs placebo group, which according to the authors supports the concept that targeting immune responses to mosquito saliva could confer cross-protection against diverse arboviruses, offering a promising pan-viral strategy to mitigate their global impact.
Overall, the results presented in this manuscript are well-written and data well-described and findings are interesting. However, there few issues that must be addressed as follow:
Major(s)
- In the Methods section, 3 Procedures: the authors stated “Participants received 0.5 mL subcutaneous injections in the upper arm on Days 1 and 22, and monitored for 30 minutes. On Day 43, mosquito feeding was performed using uninfected, starved Ae. aegypti and Ae. albopictus mosquitoes. However, no baseline sample collection was performed before the administration of the vaccine/placebo. This is critical as seroreactivity data against mosquitoes’ saliva or the recombinant proteins must be included to demonstrated that no pre-existing immunity exist in the volunteers? Please address this point.
- In Figure 7, some cellular markers are seemed to be increased in vaccinees compare to resting individuals or the placebo group, for instance, NK and dendritic cells. Increasing evidence in the arbovirus field describing the potential role of mosquito saliva in increasing virus infection, mostly dengue virus, the assumption is that “Mosquito saliva facilitates arboviral infection by taking advantage of the host's innate and adaptive immune responses to saliva” (Manning JE et al., 2018. Mosquito Saliva: The Hope for a Universal Arbovirus Vaccine? J Infect Dis. 218(1):7-15), all of this related to the increased inflammation and cell recruitment induced by mosquito-saliva components in the site of virus inoculation that results in local cell targets recruitment to facilitate virus infection and replication. How the data presented in this paper can be interpretate regarding this assumption if vaccination seems to increased main target cells to the mosquito biting sites? Please explain this point.
- In the same Figure 7, most of the cell parameters did not significantly change once compared to the mosquitos biting in the absence of vaccination. Then, how vaccination is expected to impact differently the host immune response?
- Please improve the resolution and presentation of some figures that are hard to interpretate/read because of their sizes and small letters. Few look blurry even once Zoom-in.
Author Response
All responses by authors are indicated with a “>>” and are in red font.
All changes to manuscript are indicated with use of red font to help guide assessment.
Reviewer 2
Comments and Suggestions for Authors
The following manuscript entitled “Ags-V Plus, a Mosquito Salivary Peptide Vaccine, Modulates the Human Response to Aedes Mosquito Bites” by Barningham and colleagues submitted to Vaccines, the authors assessed human immune responses to a novel vaccine, AGS-v PLUS, containing antigens derived from mosquito salivary proteins. Vaccines and placebo-injected groups were included in the study and the cutaneous response to mosquito biting was used as primary outcome. Overall, the trial identified that the adaptive immune response markers such as CD4 Th1 and CD8 T cells signatures and leukocyte influx at 48 hours post-bite suffered significant changes in vaccinees vs placebo group, which according to the authors supports the concept that targeting immune responses to mosquito saliva could confer cross-protection against diverse arboviruses, offering a promising pan-viral strategy to mitigate their global impact.
Overall, the results presented in this manuscript are well-written and data well-described and findings are interesting. However, there few issues that must be addressed as follow:
>> We thank the reviewer for their time and thoughtful comments. We believe that addressing these points has significantly strengthened the manuscript, improved its clarity and accessibility, and enhanced its overall impact.
Major(s)
In the Methods section, 3 Procedures: the authors stated “Participants received 0.5 mL subcutaneous injections in the upper arm on Days 1 and 22, and monitored for 30 minutes. On Day 43, mosquito feeding was performed using uninfected, starved Ae. aegypti and Ae. albopictus mosquitoes. However, no baseline sample collection was performed before the administration of the vaccine/placebo. This is critical as seroreactivity data against mosquitoes’ saliva or the recombinant proteins must be included to demonstrated that no pre-existing immunity exist in the volunteers? Please address this point.
>> We thank the reviewer for raising this important point regarding baseline immune status. While our current manuscript focuses on the skin transcriptomic responses following mosquito biting, the immunogenicity component of the AGS-v PLUS vaccine trial — including serological and cellular responses to the vaccine peptides — was reported previously in a separate publication ( eBioMedicine 2022;86: 104375). In that study, blood was collected both pre-vaccination, post-vaccination pre-mosquito feeding, post-vaccination post-mosquito feeding and antigen-specific antibodies (IgE, IgM and IgG) were quantified. In terms of T cell responses, those measured both vaccine antigens and to whole mosquito saliva. However, mosquito saliva is so immunogenic that made it difficult to determine if pre-existing responses existed, or if vaccination increased them.
>> Participants were recruited from Baltimore, USA, an area with established Aedes mosquito populations, meaning that most participants were likely to have had prior natural exposure to mosquito bites and, therefore, some baseline immune reactivity to saliva. As a result, using anti-saliva responses as a baseline comparator would not reliably reflect naïvety, and was not an objective of this trial. Instead, our study design used placebo-vaccinated individuals as the comparator group, representing a "real-world" control. This allowed us to determine the effect of vaccination on the host immune response to biting, without needing to account for varied and potentially undocumented pre-existing exposure to mosquito saliva across individuals.
>> We have clarified this rationale in the revised Methods and Discussion sections of the manuscript, and we have cited the previous immunogenicity paper for completeness.
In Figure 7, some cellular markers seemed to be increased in vaccinees compared to resting individuals or the placebo group, for instance, NK and dendritic cells. Increasing evidence in the arbovirus field describing the potential role of mosquito saliva in increasing virus infection, mostly dengue virus, the assumption is that “Mosquito saliva facilitates arboviral infection by taking advantage of the host's innate and adaptive immune responses to saliva” (Manning JE et al., 2018. Mosquito Saliva: The Hope for a Universal Arbovirus Vaccine? J Infect Dis. 218(1):7-15), all of this related to the increased inflammation and cell recruitment induced by mosquito-saliva components in the site of virus inoculation that results in local cell targets recruitment to facilitate virus infection and replication. How the data presented in this paper can be interpreted regarding this assumption if vaccination seems to increase main target cells to the mosquito biting sites? Please explain this point.
>> We thank the reviewer for raising this important point regarding the apparent paradox between increased leukocyte infiltration following vaccination and the possibility of enhanced viral susceptibility. While it is possible that enhanced cell recruitment alone could facilitate arboviral infection, the phenotype and activation state of recruited cells is equally critical. Our transcriptomic data suggest that AGS-v PLUS vaccination, particularly with Montanide adjuvant, skews the immune environment toward enhanced antiviral readiness, with upregulation of IFN-stimulated genes, cytotoxic markers such as granzyme B, and T-bet–driven Th1 profiles. In this context, the recruited NK and dendritic cells are likely to be in an activated, antiviral state, making them less permissive to infection and more capable of early virus control. Moreover, while the exact identity of virus-permissive skin cells in humans remains unclear, mouse models suggest that monocytes and immature dendritic cells are key targets. Our deconvolution did not show consistent increases in monocytes post-vaccination. Instead, we found that neutrophil-associated inflammatory genes, previously linked to enhanced viral susceptibility, were actually reduced in vaccinated individuals. Together, this indicates that vaccination does not simply increase cellular targets for the virus but rather reprograms the immune environment to enhance host resistance to infection. The discussion has been updated to include these nuanced but important points.
In the same Figure 7, most of the cell parameters did not significantly change once compared to the mosquitos biting in the absence of vaccination. Then, how vaccination is expected to impact differently the host immune response?
>> While it is true that many leukocyte subset signatures in Figure 7 did not show statistically significant changes with vaccination, our focus was on those key cell types that did change, particularly NK cells and dendritic cells, which are relevant for antiviral responses. It is also important to note that our study assessed a single timepoint (48 hours post-bite), chosen based on peak gene expression observed in prior studies. It is possible that other immune cell types are differentially recruited or activated at earlier or later timepoints, especially innate subsets such as neutrophils and monocytes. This is supported by work from Guerrero et al. (2022), which showed time-dependent dynamics in immune cell infiltration following mosquito bites in humans. We now mention this in the Discussion to better contextualise our findings.
Please improve the resolution and presentation of some figures that are hard to interpretate/read because of their sizes and small letters. Few look blurry even once Zoom-in.
>> We believe the reduced clarity may be due to automatic downscaling during the manuscript submission or PDF generation process. We will ensure that all final figures are provided at high resolution and meet the journal’s specifications to ensure clarity in the published version.
Round 2
Reviewer 2 Report
Comments and Suggestions for Authors
Dear Authors.
All comments and suggestions have been addressed accordingly. No more revisions are needed.
Best.